# The Association of Physical (in)Activity with Mental Health. Differences between Elder and Younger Populations: A Systematic Literature Review

**DOI:** 10.3390/ijerph18094771

**Published:** 2021-04-29

**Authors:** Laia Maynou, Helena M. Hernández-Pizarro, María Errea Rodríguez

**Affiliations:** 1Health Policy Department, London School of Economics and Political Science, London WC2A 2AE, UK; L.Maynou-Pujolras@lse.ac.uk; 2Centre for Research in Health and Economics (CRES-UPF) Mercè Rodoreda Building, Universitat Pompeu Fabra, 08005 Barcelona, Spain; 3Tecnocampus, Universitat Pompeu Fabra, 08302 Mataró, Spain; 4Freelance Researcher, 31007 Pamplona, Spain; maria.errea85@gmail.com

**Keywords:** systematic literature review, physical activity, diagnosed mental health, ageing, clinically relevant mental health

## Abstract

Background: Physical activity is associated with mental health benefits. This systematic literature review summarises extant evidence regarding this association, and explores differences observed between populations over sixty-five years and those younger than sixty-five. Methods: We reviewed articles and grey literature reporting at least one measure of physical activity and at least one mental disorder, in people of all ages. Results: From the 2263 abstracts screened, we extracted twenty-seven articles and synthesized the evidence regarding the association between physical (in)activity and one or more mental health outcome measures. We confirmed that physical activity is beneficial for mental health. However, the evidence was mostly based on self-reported physical activity and mental health measures. Only one study compared younger and elder populations, finding that increasing the level of physical activity improved mental health for middle aged and elder women (no association was observed for younger women). Studies including only the elderly found a restricted mental health improvement due to physical activity. Conclusions: We found inverse associations between levels of physical activity and mental health problems. However, more evidence regarding the effect of ageing when measuring associations between physical activity and mental health is needed. By doing so, prescription of physical activity could be more accurately targeted.

## 1. Introduction

Over a third of the world’s population is currently affected by a mental health condition, or will be during their lives [1]. A recent report from the European Union (EU) Health Programme 2014–2020 estimates that the overall one-year prevalence of mental health disorders is around 38% [2]. Indeed, these types of disorders are the third biggest cause of disability-adjusted life years (DALY) in Europe [3]. Mental health is defined by the World Health Organization as “*a state of well-being in which every individual realises his or her own potential, can cope with the normal stresses of life, can work productively and fruitfully, and is able to make a contribution to her or his community*” [4]. Several factors influence mental health. Lifestyle aspects such as physical (in)activity [5], unhealthy diets, alcohol and drug consumption [6], social context [7], work life [8], or family background [9] have been shown to impact on mental health in different contexts.

This paper focuses on the relationship between mental health (MH) and physical activity (PA). Physical activity (PA) does not only include sports and active forms of recreation (e.g., dancing), but also refers to mobility (walking and cycling), work-related activities and household tasks [5]. PA can improve physical health, self-esteem and quality of life which, in turn, enhances well-being and mental health [10]. Numerous health organisations (CDC, WHO, Health and Human Services) have outlined the benefits of physical activity, including a reduction in the risk of suffering mental health problems. Consequently, recommendations have been made on the minimum amount of activity that should be undertaken for all age groups [5]. Yet, despite the apparent benefits, 25% of all adults and 75% of teenagers (individuals aged between 11 and 17 years old) do not achieve these recommendations [5]. Physical inactivity has been defined by the WHO as a global public health problem, “partly due to people being less active during leisure time and an increase in sedentary behaviour during occupational and recreational activities” [11].

Evidence has acknowledged beneficial effects of PA on MH for the elderly [12], as well as for younger populations [13]. However, extant literature shows poor adherence rates to the prescription of PA. This non-adherence is more prominent among patients with MH [14] as well as with an increase in age [15], or for people presenting chronic diseases [16,17]. Some studies also suggested that the effect of PA on MH is stronger for elder populations than for younger adults [18]. Despite this evidence, it is rare to find papers looking at heterogeneous effects by age exploring this association between PA and MH. Additional problems found with the currently available evidence are that the studies are: (i) mostly based on self-reported measures of both PA and MH, which can lead to potential biases (e.g., [19]), (ii) when evaluating the association of PA with self-reported MH, do not analyse or distinguish respondents’ levels of self-reported MH, but treat them as a continuum (e.g., [20]); (iii) based on small sample sizes (e.g., [21]); (iv) based on cross-sectional data (e.g., [22]); or (v) purely descriptive (e.g., [23]). All of these are limitations that impede the quality of the current generation of evidence.

The aim of this paper is to explore and summarise published evidence regarding the association of PA with MH outcomes, and explore heterogeneous effects for elder and younger populations. Specific objectives are to assess whether: (i) there are differences in the association of PA with MH between the elder and younger populations; (ii) there are differences in the association of PA with MH according to the type of PA measured (objective vs. subjective); and (iii) there are differences in the association of PA with MH according to the type of MH measured (objective vs. subjective)—with a focus on clinically relevant symptoms when MH is subjective.

Hence, there are two main contributions from this review. First, we look for heterogeneous effects in the literature by age (below and above 65 years old). Second, we distinguish between objective and subjective measures of both PA and MH; moreover, subjective self-reported measures are distinguished by the use of validated scales. In addition, we have also identified that previous reviews lead to weak findings because they include papers based on (i) clinically irrelevant MH problems and (ii) descriptive non-robust statistical analysis. Thus, our goal is achieved by conducting a systematic literature review, focusing on studies conducting some type of econometric analysis, excluding studies that are purely descriptive, and selecting evidence based on clinically relevant MH problems. A clinically relevant MH problem is defined by validated score cut-offs for certain instruments used in the measurement of self-reported MH problems; e.g., a score over 10 points in the CES-D questionnaire is used as an indicator of clinically relevant depression symptoms in Ball et al. [24], according to a previous validation study [25]. As summarized in Figure 1, weak evidence is excluded, minimizing the risk of biased results. Imposing strong inclusion criteria (clinical characteristics and methodological restrictions) ensures better comparability among the selected studies, guaranteeing the robustness of our findings. Our selection criteria make it more likely that people self-reporting MH problems resemble clinically diagnosed patients than in the alterative situation where we might include all those other papers that use self-reported MH measures scores as a continuum, or that do not use a cut-off score. Additionally, restricting inclusion to papers using econometric methods means that the included papers provide information that can be used to establish an association of PA with MH, something that would not be possible using papers conducting purely descriptive analyses.

This is the first systematic review that has filtered these analyses identifying the specific association of these practices with MH for the population that either has a clinical diagnosis of MH or has clinically relevant symptoms of MH (objectively measured).

The paper is organised as follows. In Section 2 we present the methodology, in Section 3, the results, Section 4 presents the discussion, and finally, Section 5, concludes.

## 2. Material and Methods

This systematic review followed the Preferred Reporting Items for Systematic Reviews and Meta-Analyses (PRISMA) statement [26]. The framework of this systematic review according to PICO [27] was: Population: people with mental health disorders, either diagnosed, or clinically relevant when self-reported; Intervention: Physical Activity of any type, objective or self-reported; Comparison: Elder and younger populations; Outcomes: effect of physical activity over mental health.

### 2.1. Search Strategy

We conducted our search using PubMed/Medline and EconLit as our main databases for this systematic literature review. Other sources were also consulted to complete the search with papers that were identified after reviewing some of the included records.

We applied the PICO/PECO method to structure [28] and combine keywords regarding MH (that included “Mental health”, “Mental disorder”, “Depress*”, “Anxiety”, “Psychiatr*”), as well as keywords for PA (“Physical activity”, “physical inactivity”, “Physical exertion”, “exercise”, “physical exercise”, “sport”, and “physical education”) and econometric methods (“Quantitative studies”, “quantitative analys*”, “regression”, “econometric*”, “association”, “cross-section*”, “longitudinal analys*”, “panel data analys*”, “causality*”). We excluded descriptive studies (using the words “descriptive analys*”, “ANOVA”, and “correlations measures”).

We combined these words using an algorithm and the boolean terms OR, AND, and NOT. Our PubMed/Medline and EconLit search strategy, focused on finding those records containing the following terms in titles or abstracts, is provided as Appendix A.

The search strategy for PubMed/Medline, EconLit and additional sources is presented in Figure 2 below. Reference lists of primary research reports were cross-checked in an attempt to identify additional studies.

### 2.2. Eligibility Criteria

We limited records to any academic articles or grey literature published since 2000 available in full-text format, assessing the association of PA (whether this was objectively or subjectively measured) with MH (objectively measured, or population has at least clinically relevant symptoms). We sought studies that used econometric analysis methods (i.e., regression analysis) to establish an association of PA with MH. Papers with a different objective, and purely descriptive papers—even if they were pursuing this objective—were excluded. Studies were also excluded when investigating only symptoms of mental disorders. We did not filter for age groups in order to capture publications for all age groups, allowing comparisons by age groups. Meta-analyses, systematic reviews, methodological papers, congress proceedings, meeting abstracts and case studies were excluded from the search. We also excluded papers that presented a high risk-of-bias. All identified reasons for exclusion are detailed in the PRISMA flow diagram (Figure 2).

We consider objective and subjective measures for both PA and MH (if subjective, only those measures that are clinically relevant). Objective measures of PA are those recorded by an external technology (e.g., accelerometer recording number of steps or time spent performing the exercise) or by an exercise supervisor (e.g., coach). PA that is manually reported by the individual, for example, through a questionnaire or interview, is considered a self-reported type of PA. Regarding MH, measures are considered self-reported MH when they are not measured through a medical diagnosis. Only medical diagnoses are considered objective measures of MH. Subjective measures considered for PA and MH can be distinguished between those measured based on validated scales (e.g., IPAQ questionnaire, the only validated scale found for PA in this review, or GHQ-12 or PHQ-8, amongst others, for MH) and non-validated scales (e.g., questions for PA such as “How often are you physically active or perform exercise during your leisure time? (excluding domestic work)”), and questions for MH such as “Have you ever been diagnosed with depression?” Note that there exist other validated scales for measuring objective PA, such as the GPAQ questionnaire. However, there were no studies using the GPAQ questionnaire that satisfied the inclusion criteria specified for the objective of this review.

### 2.3. Study Selection and Data Extraction

After completing the search in each database, all references were imported into Zotero, the bibliographic software programme in which the study selection was conducted. The study selection included the screening of titles and abstracts in a first stage, and full-texts in a second stage, conducting a forward and backward search. The search and study selection were conducted in January 2021 by two researchers (M.E and L.M) independently from each other. Any doubts or disagreements between the two researchers were discussed with a third researcher (H.M.H.-P.). The methodology followed for data extraction was reviewed and approved by all authors. It was not necessary to contact any of the authors of the papers included in this review for completion of missing relevant information from the article.

### 2.4. Risk of Bias Assessment

We followed the method developed by Parmar et al. [29] for assessing the risk-of-bias of our included records. This includes seven key domains: selection bias, ecological fallacy, confounding bias, reporting bias, time bias, measurement error in exposure indicator, and measurement error in health outcome. For each publication, we rated each of the abovementioned domains: a score of 1 is given for a low risk of bias, 2 for a moderate risk and 3 for a high risk. Then, we computed the overall rating as follows: 1 (strong) was given if none of its domains were rated as weak, 2 (moderate) if up to two domains were rated as weak, or 3 (weak) if three or more domains were rated as weak.

### 2.5. Synthesis of Results

Data extraction from the selected papers focused on the following fields: authors and year of publication, type of study (RCT, cohort with follow-up, cross-sectional), study’s objective, sample size (and % of MH patients), age range (and mean age of the study sample), PA measure (self-reported (validated scale or not)/objective (programme)), MH problem assessed, MH Patient reported outcome (PRO) measure (self-reported (validated scale or not)/objective (clinical diagnosis)), results of the study (regarding the association of PA with MH only -all other results unrelated with these objective were not extracted-), and overall effect found for the association of PA with MH. These fields were used to construct our summary result table (Table 1).

Next, we classified studies in clusters according to the different criteria categories: age (we used 3 categories: all ages, <65 and 65+), PA type of measure (3 categories: objective, subjective validated scale, and subjective non-validated scale), and MH type of measure (3 categories: objective, subjective validated scale, and subjective non-validated scale). Thus, we ended up with 27 potential clusters. The result of this classification is summarised in Table 1 and in the Main Results section.

We limited our synthesis to studies that reported results of the association of PA with MH for a minimum sample size in each group of individuals. Our minimum study sample size requirements were (i) a minimum of 10% individuals from the total study sample with self-reported/diagnosed MH in samples with less than 100 individuals, or (ii) in samples of more than 100 individuals, a minimum of 5% of individuals with self-reported/diagnosed MH. Moreover, population-based studies representative of the general population were preferred. Alternatively, a minimum power of 0.80 and significance of 0.05 were required for a study to be able to detect group differences. Studies needed to report estimates, *p*-values and 95% confidence intervals from the econometric model. We only considered those studies that were moderate or strong at the quality and risk of bias assessment. The weakest studies were excluded.

We considered an association of PA with MH existed when the paper showed significant results in the econometric analyses (*p*-value < 0.05). When a paper explored the association of PA with MH for different population subgroups (e.g., age groups), when available, we extracted the specific information for the overall population as well as the information on the association found for each subgroup. We considered there was no association when the paper reported that no effect of PA on MH was found, or when the differences found were not statistically significant.

Finally, we summarised and interpreted our analysis according to the available combinations of PA and MH measures (PA and MH objective and/or subjective), and for the different age groups identified. This ensured the provision of the most complete interpretation of the selected studies’ results for this review.

## 3. Results

### 3.1. Study Selection

Our search strategy identified 2268 potential studies from PubMed (2249), EconLit (13) and other sources (6). After removing duplicates, 2263 abstracts remained for title and abstract screening. We excluded 2125 abstracts and selected 138 for full-text screening. Among these, only 1 paper was selected from the EconLit search, 136 were selected from the search at PubMed, and 1 paper was manually retrieved from Google scholar given our awareness of the study and its relevance. We finally extracted information from 29 papers, and excluded 2 papers after performing the risk-of-bias check. A Prisma 2009 flow diagram representing the study selection process has been presented in Figure 1.

Table 1 summarises the characteristics of the 27 studies finally included in this systematic review. Table 2 summarises the studies’ objectives and their results, including a column with the overall effect (positive, negative or none) that can be concluded after reviewing each study.

The majority of the reviewed studies found a negative association of PA with MH, and positive for physical inactivity studies. There are two studies [30,31] that found no association of PA with MH, and a third that found association with depression symptoms, but no association for patients with anxiety [32].

### 3.2. Quality Assessment and the Risk of Bias

We use the Parmar et al. [29] scale for risk of bias and quality assessment. We evaluate the qualities of all included studies in the qualitative synthesis, based on a set of seven questions. Some of these questions needed to be adapted for this paper. For example, for RCTs, because representativity does not apply, we evaluate selection bias by analysing the appropriateness of the sampling methods (e.g., the study reports good power of the sampling methods reported, large-enough sample sizes, blinding, randomisation methods). Regarding confounding bias assessment, we consider stronger those studies that included an indicator of individuals taking a MH treatment as a control variable. For time bias, we consider that the longer the distance (in years) between the timeframe analysed and the time of publication, the higher the risk of time bias. A higher risk of measurement error in the exposure variable or in the MH measurement is assumed for self-reported types of PA and MH, respectively, especially when the PA/MH measurement instrument used was not a validated scale.

Each item of the bias scale was rated into one of three categories according to its risk of bias—low (for strong studies with little risk of bias), moderate, or high (weak studies with high risk of bias). When converting these elements into an overall bias score for each paper, the overall assessment of two studies was “weak”, and these were automatically excluded for the review (not shown in the summary of studies in Table 1). Among the included studies, there were fifteen strong studies (having no “weak” ratings) and twelve of moderate quality (with maximum one “weak” rating).

### 3.3. Main Results

Of the 27 studies included in this review, 14 (51.8%) were cross-sectional studies, 2 RCTs, and 11 follow-ups of a cohort. Different PA measures were assessed: 6 programmes of PA reporting an objective measure of PA [30,31,36,44,46,47], and the remaining 21 offering conclusions from studies based on self-reported PA. The included studies were critically assessed and their main focus was on finding an association of different types of PA with objective/clinically relevant symptoms for MH [31,39,41,46,47]. We identified in Table 1 the number of studies that used objective or self-reported, and validated vs. not validated, measures of PA and MH. Most studies used self-reported validated MH measures (85%), while for PA 48% used non-validated self-reported measures, 30% used validated measures and 22% used objective measures. There were only 4 studies presenting results for the elderly, and they all used validated self-reported MH measures. There was only a minority of 5 studies that, when measuring the association of PA with MH, accounted (as a confounding factor) for some type of MH treatment [31,39,41,46,47].

#### 3.3.1. Differences in the Association of PA with MH between Elder and Younger Populations

Among the 27 studies included, 14 included elder populations of 65 years or over [31,32,33,34,35,36,37,39,42,44,45,51,54]. However, only 2 studies [17,40] included as covariates the interaction between age groups and PA to facilitate the comparison across age groups of PA on MH. In particular, Griffiths et al. [40] found a lower risk of mental ill-health for mid-life (AOR = 0.81 (0.66–0.99) for ≥60 MET hours/week) and older women (OR = 0.77 (0.55–1.07) for ≥60 MET hours/week) who reported increased levels of physical activity than those who did not increase physical activity.

There were 9 studies that [32,33,35,36,39,44,49,51,54] included age as a control variable, but the analysis was performed in a way that did not allow any conclusions to be made regarding the differences in the association of PA with MH between the elder and the younger populations. One study, Hamer et al. [18], observed slightly stronger associations of PA with self-reported MH in participants >60years of age or with chronic conditions.

The study of Bishwajit et al. [35] is based on self-reported PA while Karg et al. [46] is based on objectively measured PA (programme). Bishwajit et al. [35] found, for the population aged 50 and older (mean age ~ 60, SD ~9), that those who reported never engaging in self-reported moderate or vigorous PA had ORs clearly higher for diagnosed depression than those who engaged in moderate or vigorous PA. Karg et al. [46], who focused on a middle-aged population (mean age = 42, SD = 12.5), found support for previous findings suggesting positive effects of physical activity and particularly bouldering in depressed individuals. This study also controlled for participants’ current therapeutic treatment, in addition to the prescribed PA. Comparing ORs for both studies, the effect is higher in the bouldering therapy programme [46]. One should take into account that their population is also slightly younger.

#### 3.3.2. Differences in the Association of PA with MH between Self-Reported and Objective Types of MH

There were only 2 studies out of the 27 that included a population with a clinically diagnosed mental health disorder [35,46]. The remaining twenty-five studies assessed self-reported mental health using validated scales, with the most frequent the GHQ-12 (n = 4 studies) ahead of the GDS-15 (n = 3), CES-D (n = 3), PHQ-9 (n = 2), and the SF-36 (n = 2). All of the papers selected for this review which analysed the association of PA with MH based on self-reported MH used a MH cut-off score, meaning their MH measures should be considered as the probable presence of a MH problem. As is common in the literature, when an individual scores above that cut-off, this individual was considered to have clinically relevant symptoms of a MH disorder. We also included 2 studies [38,53] that, even though they used a validated MH scale, did not use a specific cut-off but rather performed analysis by categories of severity of symptoms.

Among the 25 studies based on populations’ clinically relevant symptoms of MH as identified by self-reported mental health measures, 18 studies concluded that there was an unconditional, negative association between PA levels and MH prevalence, fifteen of them indicating PA is beneficial for MH, and three indicating physical inactivity worsens MH; 1 study reported differences between PA but only for depression, not for anxiety [32]; 1 study found that PA is especially beneficial for the MH of the elder population. Finally, 1 study found a beneficial but only for women [31]. Two studies did not find an association of PA with MH [30,31].

#### 3.3.3. Differences in the Association of PA with MH for Self-Reported and Objective Types of PA

Among the 27 studies reviewed and analysed, four assessed impact on MH with an objective measure of PA [30,31,36,46]. Objective measures of PA included supervised exercise programmes, accelerometer/activity monitor, and bouldering psychotherapy. The majority of studies (N = 21, 77.7%) assessed the association with MH using a self-reported measure of PA. The most repeated instrument for self-recording PA time was the IPAQ questionnaire (n = 3), while all other studies used different questions to assess time dedicated to PA. Among the studies using a self-reported measure of PA, 3 assessed physical inactivity [44,49,50], and all them found it was associated with adverse MH.

Of the 4 studies using objective PA, 2 found that higher PA was associated with lower levels of poor MH [36,46], and 2 found no effect, one of which studied an elderly population [31] and the other a population of post-partum women [30]. Within the self-reported PA measures, more PA led to better MH in 18 studies, and more physical inactivity led to worse MH in three studies. Fourteen studies conclude that this effect of PA is persistent without restrictions. A similar effect was identified but with some restrictions, for PA, in four studies. Some techniques were found to be more effective than others [37], or MH might be effective for depression but not for anxiety [29], or it showed effectiveness especially on a subgroup of the population (e.g., aged > 60 or with chronic conditions, as in one of the studies [18], or for women only [33]). Two studies conclude there was no association of PA with MH [30,31].

## 4. Discussion

This systematic review aimed to present and rigorously assess the evidence available on the association of PA with MH and differences by (i) age groups (elder and younger populations), (ii) type of MH (self-reported and objectively measured), and (iii) type of PA measure (objective vs. self-reported) in order to identify literature gaps, document the current leading-edge knowledge, and open a discussion regarding the direction in which further research should move. Our review results indicate that physical activity is beneficial for mental health. However, the evidence was mostly based on self-reported physical activity and mental health measures, and did not allow to really compare results between younger adults and adults aged 65 or over.

Given the number of abstracts captured by our search strategy, one could think that there exists an extensive literature on the association between PA and MH outcomes. However, a large number of studies were excluded (N = 65 excluded records at the full-text screening phase, representing 47.1% of all the full-text screened records, as stated in Figure 1) because they included in their analytic samples individuals who had low MH symptoms as well as those with probable MH issues [55], despite the differences between these two populations. Failing to account for this weakness reduces the validity and precision of previous reviews.

Imposing this strong inclusion criterion is based on medical literature. There is evidence suggesting people clinically diagnosed with MH and people who self-report to be suffering from MH are very different [56,57]. In addition, despite the validity of the instruments that could be used to identify self-reporting people with clinically relevant symptoms of MH, most published papers ignore the fact that these two populations (reaching or not the cut-off) are different, and treat them without making a distinction (e.g., [58,59,60] and many others). Different cut-offs are recommended, specific for each scale or instrument, to assess the severity of MH disorders, and to distinguish people who would be very likely to be diagnosed with a MH disorder from those with less severe MH symptoms. Although for some instruments the cut-off points are still unclear [61], there are now many instruments that have been validated and for which high degrees of sensitivity have been demonstrated [62,63]. In consequence, studies like Zang et al. [54] have demonstrated significantly different associations of PA with MH for individuals with self-reported but not-clinically relevant symptoms, and for those with clinically relevant symptoms. In spite of evidence to support the validated cutoffs used to screen for MH problems linked to clinically relevant symptoms [57], the papers we exclude from our study ignore them. Instead, they treat MH outcome scores as continuous variables when exploring the effect of PA on self-reported MH, or on mental health scores that are not confirmed by a clinician [64]. These studies group all of the participants who self-report an MH problem in the same category as those with a clinical diagnosis of MH, which is imprecise and weak.

Literature has also found consistently that moderate-to-vigorous intensity physical activity improves MH of the mentally-ill [35,36,37,65]. Physical activity could, indeed, be an effective measure for both preventing and treating MH. While psychotropic medications are still the main treatment for most MH disorders [66] a growing body of scientific evidence strongly supports the role of exercise in the treatment regime [67]. For example, Zhang and Yen [54] used econometric models to demonstrate that physical activity remedies the depressive symptoms amongst individuals suffering from mild and moderate depression. Although Lordan et al. [68] confirmed these results and added that the impact is even greater for women, their study was based upon a population with MH symptoms, with no screening indicator for the clinical relevance of such symptoms. Physical activity remedying depressive symptoms has also been analysed through different categories such as green spaces, group exercise, the elderly, youth, gender and countries/regions. However, the association of these with populations suffering from MH problems, or with, at least, probable MH problems is still uncertain.

This is the first systematic review, to our knowledge, assessing the association of PA with MH combining studies of populations with diagnosed MH and those with self-reported MH and clinically relevant symptoms. We believe including the subsample of people with probable MH is important given their proximity to MH diagnosis.

Indeed, if we compare the results observed in those studies using patients clinically diagnosed with MH against results from studies using self-reported clinically relevant MH measures, we observe similar findings. In particular, the two studies using populations of patients with a MH diagnosis, and twenty-one out of the twenty-five studies using self-reported clinically relevant MH measures, found a negative and unconditional association between levels of PA with levels of MH, indicating that PA is beneficial for MH (and (in)PA worsens MH). The remaining four studies using self-reported MH measures found a conditional association, for example, that some therapies would work better than others to help patients with MH.

In addition to this, we designed our study selection and search strategy to ensure that we captured populations of all ages within our studies. Therefore, we were able to use age as a comparison factor, putting our focus on the differences of the association of PA with MH between elder and younger populations. Although our review provides evidence indicating PA is beneficial for MH, we observe that the intensity of such a relationship varies by the type of PA and MH measured, as well as by age. The number of studies offering a cut-off distinguishing clinically relevant symptoms from less important symptoms of MH is small compared to the number of studies that use MH self-reported outcome measures without making this distinction. These findings suggest that more evidence is needed regarding (1) the association of physical activities with mental health for people of different ages; and (2) in people with probable MH, ignoring the similarity between this population and the population with a clinical diagnosis on MH. Another gap we identify in the literature is the lack of longitudinal studies, with most studies analysing cross-sectional data or short-term follow-ups. This creates a barrier to establishing causality in the analyses of the association between PA and MH. We, thus encourage further studies to use validated cut-offs to provide analyses of people with possible MH in the future.

Our paper has some limitations. First, we did not include papers published before 2000. However, the time constraint decision lies on the fact that most of the MH measures including a cut-off to distinguish for clinically relevant symptoms were validated after the year 2000. Hence, we considered that a search focused on 2000 and onwards papers would be more accurate. Second, our analysis is not purely based on clinically diagnosed MH, the most accurate measure of MH, given the low number of published studies with clinically diagnosed populations (n = 2). Thus, we also included papers based on population with clinically relevant symptoms of MH. Yet, these clinically relevant symptom cut-offs were created specifically to indicate the high probability of individuals to be diagnosed in the early future, which mitigates, somewhat, this limitation.

## 5. Conclusions

We found inverse associations between PA and MH. However, research designs are often weak, based mostly on self-reported measures of PA and MH, and effects are small to moderate. Effect by age seems to be scarce when measuring the differences in the association of PA with MH. More studies are required to provide an accurate estimate of the association of PA with MH, using more robust methods which can be externally verified for different populations. In order to better target and effectively prescribe PA, more evidence comparing elder and younger populations, and the specific populations with probable MH, is required. 

## Figures and Tables

**Figure 1 ijerph-18-04771-f001:**
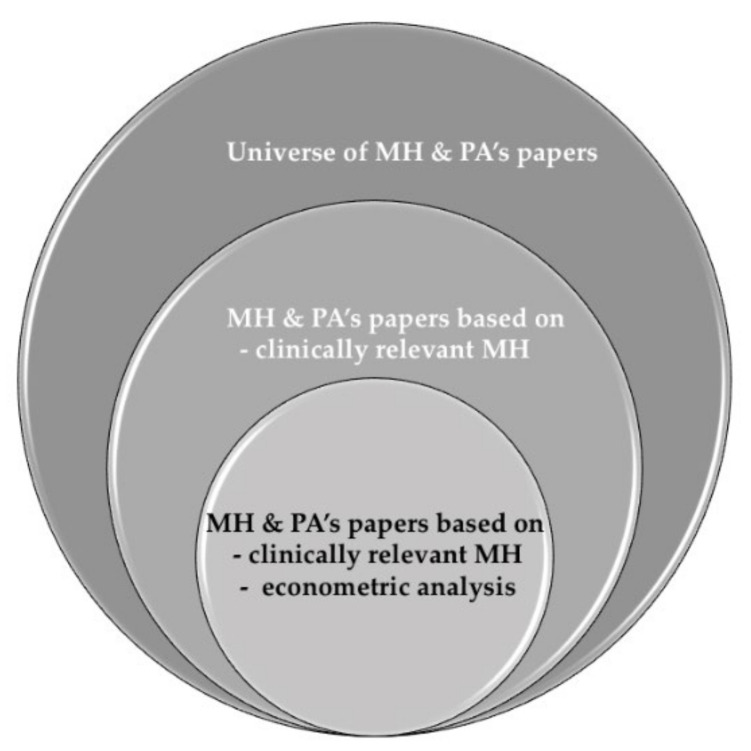
Evidence on MH and PA.

**Figure 2 ijerph-18-04771-f002:**
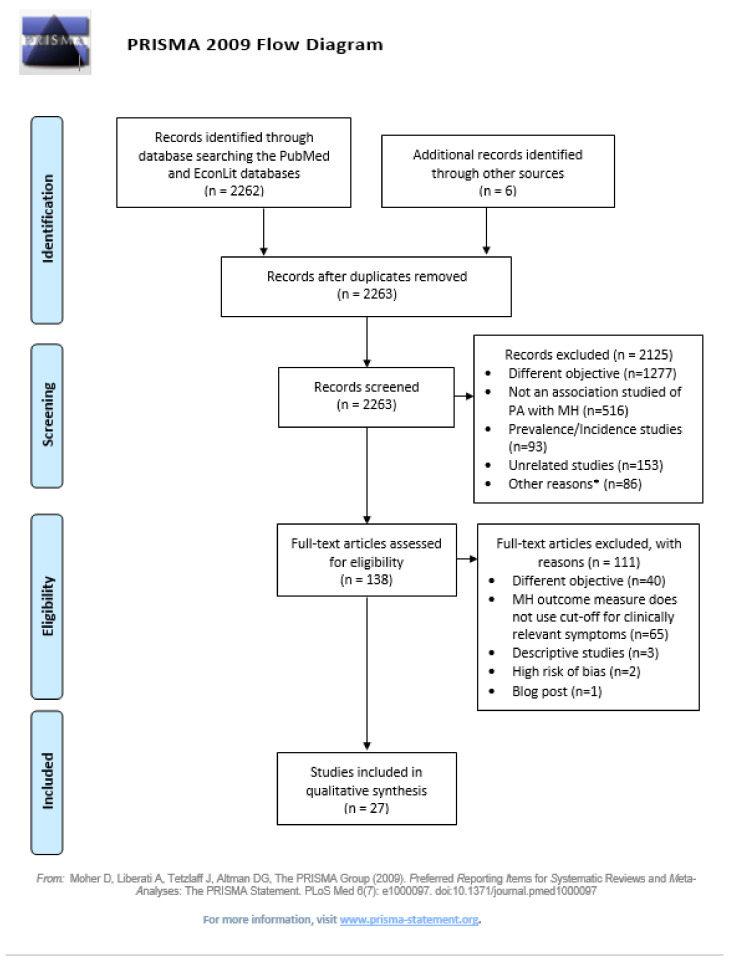
PRISMA flow diagram. * Other reasons for exclusion at the eligibility stage included: systematic reviews or meta-analyses (n = 48), studies of small sample size (n = 9), studies that were work in progress (n = 8), guidelines (n = 6), indirect influence of PA with MH measured only (n = 4), pilot studies (n = 3), MH outcome does not use cut-off for clinically relevant symptoms (n = 2), descriptive studies (n = 1), associations based on beliefs (n = 1), convenience sample (n = 1), not in English or Spanish (n = 1), qualitative studies (n = 1), and specific population with risk of selection bias (n = 1).

**Table 1 ijerph-18-04771-t001:** Characteristics and summary findings of the included studies.

Author (Year) Study Name (Reference) ACRONYM	Type of Study	Study Objectives	Sample Size (MH Sample, %)	Age Range, Mean (SD/IQ Range)	PA Measure (Self-Reported/Programme)	MH Problem(s) Assessed	MH PRO Measure (Objective/Self-Reported) Cut-Off for CRS	Results	Overall Effect (+/−, No)
Annerstedt et al. (2012) [33] Health survey ¥	Cohort (follow-up)	Whether or not an inverse relationship exists between green qualities in the neighbourhood and development of mental disorder.	8683 (1424, 16.4%)	18–80 50.1 (unknown)	“How often are you physically active or perform exercise during your leisure time? (self-reported) Not validated measure	Poor Mental Health	GHQ—12 (self-reported) Reporting problems in 3 or more questions Validated measure	The risk of having poor mental health at follow-up decreased 80% if having access to space and being physically active and 70% if access to Space and physically active, compared to not having access to either of these qualities and being physically inactive. These effects were statistically significant for women, but not for men. However, the tendencies were the same for men. We have found that in interaction with physical activity the qualities Serene and Space have some risk-reducing effect on mental health disorders for women, an impact that seems to over shadow the mere amount of nature.	- (conditional)
Ball et al. (2009) [24] The Australian Longitudinal Study on Women’s Health/ALSWH ¥	Cohort (follow-up)	To examine, in a population-based sample of young women, the prospective associations over 3 years between (i) BMI and PA, and depressive symptoms; and (ii) changes in BMI and PA, and depressive symptoms.	6677 (1691, 26%)	21–65 Mean not disclosed	Frequency and duration of walking (for recreation or transport), and of moderate- and vigorous-intensity activity in the last week. (self-reported) Not validated measure	Depression	CES-D (self-reported) Score ≥ 10 Validated measure	Adjusted odds of depressive symptoms in 2003 were lower among women who reported any level of PA, compared with women who reported none. After adjustment for sociodemographic variables and BMI, ORs for depressive symptoms in 2003 became nonsignificant for the very low category, but remained significantly lower among women who reported low, moderate (borderline significant or high levels of PA. Compared with women who maintained none or a very low level of PA, those who increased their PA level from none or very low to either a moderate or high level had significantly lower risk of depressive symptoms in 2003, which remained after adjustment for covariates and 2000 BMI, and also after adjustment for covariates and change in BMI (latter data not shown).	-
Benedetti et al. (2008) [34] Survey Θ, *	Cross-sectional	To evaluate the association between physical activity level and mental health status among elderly people.	869 (291, 33.5%)	60–101 71.6 (SD = 7.9)	International Physical Activity Questionnaire (IPAQ) (self-reported) Validated measure	Dementia and Depression	GMS scale (self-reported) Dementia: score ≥ 3 Depression: score ≥ 7 Validated measure	The present study found a significant relationship between the levels of physical activity and the state of mental health. That is, this association showed lower prevalence of indicators of depression and dementia among non-sedentary elderly people. The importance of keeping active was reaffirmed, along with the fact that physical activity influences how depressive syndrome is faced, through expanding sociability and corporal stimulation. It can be inferred that physical activity is able to reduce and/or delay the risks of dementia, although it cannot be stated that physical activity avoids dementia.	-
Bishwajit et al. (2017) [35] ¥	Cross-sectional	To explore the pattern of physical activity across various demographic and socioeconomic groups in four countries, and to measure the association between PA and self-reported depression among the middle- and older-aged population	6855 (2380, Prevalence of self-reported depression was respectively 47.7%, 40.3%, 40.4% and 11.4% in Bangladesh, India, Nepal and Sri Lanka respectively)	50 and over Bangladesh [60.72 (SD = 9.6)]; India [59.94 (SD = 9.1)]; Nepal [60.5 (SD = 9)]; Sri Lanka [60 (SD = 9.08)]	Time dedicated to MPA and VPA (self-reported) Not validated measure	Depression	Clinical diagnosis (objective)	With regard to diagnosed depression, compared to those who reported engaging in MPA on daily basis, the odds of reporting depression were more than five times higher [AOR = 5.512; 95% CI = 1.159–26.21] for those who never took MPA in Bangladesh. In India, those never took VPA had 44% higher [AOR = 1.442; 95% CI = 1.046–1.987] odds of being diagnosed with depression compared those who never engaged in VPA.	-
Blumenthal et al. (2012) [36] HF-Action ¥	RCT	To determine whether exercise training will result in greater improvements in depressive symptoms compared with usual care among patients with heart failure.	2322 (653, 28,1%)	19–91 56 (IQ range = 50–63)	Supervised & home-based aerobic exercise training sessions (programme)	Depression	BDI-II scale (self-reported) score ≥ 14 Validated measure	Exercise training may be effective in reducing depressive symptoms and by further documenting the prognostic significance of depression in patients with heart failure. In this ancillary study from the HFACTION trial, patients with heart failure who participate in exercise training, compared with usual care, had modest reductions in depressive symptoms at 12 months, although the clinical significance of these small improvements is unknown.	-
Byeon et al. (2019) [37] The Korea National Health and Nutrition Examination Survey (KNHANES) Θ	Cross-sectional	To investigate the relationship between physical activity and depression in the elderly living alone and to provide basic data for the prevention of depression in the elderly.	256 (45, 19%)	65 and over	Regular physical activities performed on average in a week were investigated using a self-care questionnaire. (self-reported) Not validated measure	Depression	PHQ-9 scale (self-reported) score ≥ 10 Validated measure	This study showed that regular flexibility exercises were independently related to depression prevention. The flexibility exercise of the elderly was independently associated with depression prevention. The results of this study implied that persistent flexibility exercise (e.g., stretching and freehand exercise) might be more effective to maintain a healthy mental status than muscular strength exercise. A longitudinal study is required to prove the causal relationship between physical activity and depression in old age.	- (some techniques more than others)
Chang et al. (2020) [38] Survey in Taiwan ¥	Cross-sectional	To investigate the long-term association between midlife PA and late-life depressive symptoms, on average 25 years later, in a population free of clinical history of depression and diagnosis of dementia.	1114 (MH sample not reported)	Range not disclosed 35.9 (SD = 15.16)	Exercise. Any activity they chose to do as their exercise (e.g., workouts at home, running outside, etc.). (self-reported) Not validated measure	Mood state	POMS (self-reported) At least 10 items answered Validated measure	There was a significant main effect of exercise frequency during the pandemic on mood states. Those who exercised four days or more had significantly higher mood states compared to those who exercised for 2–3 days (bduring3-2 = 0.14, *p* = 0.04), and those exercised for 2–3 days had significantly higher mood states compared to those who exercised one day or less per week during the pandemic (bduring2-1 = 0.29, *p* < 0.001). There was also a significant main effect of pre-pandemic exercise frequency on mood states. Specifically, those who exercised four days or more per week pre-pandemic had a significantly lower mood state during the pandemic, compared to those who exercised for 2–3 days per week pre-pandemic (bpre3-2 = 0.16, *p* = 0.03). However, there was a significant interaction effect on exercise frequency levels during the pandemic x pre-pandemic exercise frequency levels on mood (bpre x during = 0.48–0.42, *p* = 0.01–0.03). Meaning, the effects of pre-pandemic exercise frequency on mood were dependent on exercise frequency during the pandemic.	-
Chang et al. (2016) [39] Age Gene/Environment Susceptibility (AGES)—Reykjavik Study §, Θ	Cohort (follow-up)	To investigate the long-term association between midlife PA and late-life depressive symptoms, on average 25 years later, in a population free of clinical history of depression and diagnosis of dementia.	4140 (216, 5.5%)	65 and over Mean not disclosed	Regular participation in sports or exercise, and hours per week (self-reported) Not validated measure	Depression	GDS-15 (self-reported) score ≥ 6 Validated measure	This longitudinal study over a period of 25 years found a strong association between midlife PA and depressive symptoms in late life among community dwelling old people who did not have a history of depression. Compared with those who were inactive at midlife, those who were active at midlife had significantly less depressive symptoms 25 years later even after controlling for demographics, physiological markers, and various aspects of cognitive function.	-
Chen et al. (2010) [40] Questionnaire ¥	Cohort (follow-up)	To examine the association of lifestyle factors and supplement use with depression among breast cancer survivors.	1399 (176, 12.5%)	Adults, not specified 53.7 (SD = 9.8)	Exercise questionnaire (self-reported) Validated measure	Depression (self-reported)	CES-D scale (self-reported) score ≥ 16 Validated measure	Regular exercise participation may play an important role in the prevention of depression among breast cancer survivors.	-
Coll et al. (2019) [30] The Physical Activity for Mothers Enrolled in Longitudinal Analysis (PAMELA) study ¥	RCT	To assess the efficacy of a 16-week exercise intervention during pregnancy on the prevention of postpartum depression using data from a large RCT.	639 (579, 90.6%)	Young women, range unspecified 27.1 (SD = 5.1)	16-week supervised exercise program including aerobic and resistance training delivered in 60-min sessions 3 times per week (programme)	Depression (postpartum)	EPDS (self-reported) score ≥ 12 Validated measure	There were no significant differences between study groups in the rates of postpartum depression (12 of 192 [6.3%] in the intervention group and 36 of 387 [9.3%] in the control group; OR, 0.65; 95%CI, 0.33–1.28). Sensitivity analysis using multiple imputation to deal with missing data yielded virtually identical results.	No
Feng et al. (2014) [41] Questionnaire §, ¥	Cross-sectional	To investigate the independent and interactive associations of physical activity (PA) and screen time (ST) with depression, anxiety and sleep quality among Chinese college students.	1106 (201, 18.2%)	16–24 18.9 (SD = 0.9)	Frequency of physical activity (self-reported) Not validated measure	Depression and Anxiety	SDS and SAS (self-reported) Anxiety: score ≥ 50 Depression: score ≥ 53 Validated measure	The present study suggests an independent and interactive relationship of high PA and low ST with significantly reduced prevalence of mental health problems and favorable sleep quality among Chinese college freshmen. These results provide support for the notion that maintaining sufficient PA and reducing sedentary behaviors should be included in the planning of health promotion strategies.	-
Griffiths et al. (2014) [42] Survey data derived from the Finnish Public Sector Study (FPSS) ¥	Cohort (follow-up)	To explore the relationship between physical activity and symptoms of mental ill-health in a large, well defined and heterogeneous sample of working women.	26,913 (4666, 17%)	18–69 45.6 (SD = 9.8)	Average time spent on physical activity (self-reported) Not validated measure	Mental ill-health	GHQ-12 (self-reported) score ≥ 4 Validated measure	The results of this study with a large cohort of Finnish working women showed that physical activity was associated with a reduced future risk of mental ill-health. These findings also demonstrated an inverse dose–response relationship between physical activity and likelihood of later symptoms of mental ill-health. In addition, our findings revealed that mid-life and older women who reported increased levels of physical activity were at significantly less risk of later mental ill-health than those who did not increase physical activity. No association was observed in the group of younger women	-
Guddal et al. (2019) [43] Young HUNT3 study ¥	Cross-sectional	To describe PA levels and sport participation in a population-based sample of adolescents, and to explore how they relate to mental health in different age groups.	7619 (933, 12.2%)	13–19 15.8 (SD = 1.7)	Leisure time PA and type and frequency of sport participation (self-reported) Validated measure	Psychological distress	SCL-5 (self-reported) score ≥ 2 Validated measure	In this population-based sample of adolescents, PA levels and participation rates in sports were lower among girls, and lower among senior high school students compared with junior high school students. These results showed that higher levels of PA were favorably associated with self-esteem and life satisfaction throughout adolescence, as well as with reduced likelihood of psychological distress in senior high school students. Team sport participation was associated with mental health benefits, especially for girls.	-
Hamer et al. (2017) [18] The Health Survey for England (HSE) and the Scottish Health Survey (SHS) ¥	Cross-sectional	To compare associations between objectively assessed and self-reported sedentary time with mental health in adults.	108,011 (15,661, 14.5%)	Adults, range not specified 47 (SD = 17)	Questionnaire to enquire about frequency, duration and pace of walking and participation in sports and exercises including cycling, swimming, running, football, rugby, tennis and squash (self-reported) Validated measure	Psychological distress	GHQ-12 (self-reported) Score > 3 Validated measure	The pattern of results was essentially the same in men and women and across different age categories. Slightly stronger associations were observed in participants >60 yrs. of age. Significant interaction (*p* < 0.05) by longstanding illness was observed. Results suggest that presence of chronic illness is an important factor in modifying associations between PA and mental health; among participants reporting longstanding health conditions, reduced odds of psychological distress below the PA guidelines were observed, from as little as one to two sessions per week of MVPA. Given that just under half (~44%) of this general population sample of adults reported a longstanding health condition, this is an important factor in potentially modifying associations between PA and mental health.	- (specially for the aged >60 or with chronic conditions)
Hamer et al. (2014) [44] HSE ¥	Cross-sectional	To explore if mental health benefits can be optimized by accumulating PA in certain patterns.	11,658 (1486, 12.7%)	16–95 50 (SD unknown)	Uniaxial accelerometer that records movement on the vertical axis, the Actigraph GT1M (Actigraph, Pensacola, Florida, USA), during waking hours for seven consecutive days. (programme)	Psychological distress	GHQ-12 (self-reported) score ≥ 4 Validated measure	Sedentary time is associated with adverse mental health. Sedentary time (<200 CPM) was directly associated with psychological distress after adjustment for all covariables including MVPA, although this was more apparent in the highest tertile (OR = 1.74, 95% CI 1.07 to 2.83). Light activity (200–2018 CPM) was inversely associated with risk for psychological distress, although the association was not linear. MVPA, however, was not associated with psychological distress in any models. MVPA was inversely associated with risk of psychological distress in a dose–response manner (*p*<0.001 for all models).	+
Kanamori et al. (2018) [45] The JAGES longitudinal study Θ	Cohort (follow-up)	To examine (1) the relationship between frequency of exercise at baseline and later depression in older Japanese adults and (2) the relationship between exercise patterns at baseline (non-exercisers, exercising alone only, or exercising with others) and later depression, and (3) the relationship between combinations of frequency of exercise and exercise patterns at baseline with later depression.	1422 (MH subsample not specified)	65 and over 72.5 (SD = 4.9)	Total frequency/pattern of exercise (self-reported) Not validated measure	Depression	GDS-15 (self-reported) score ≥ 5 Validated measure	The results of the present study suggest that exercising two or more times a week and/or exercising with others can lower the risk of depression in older Japanese adults. When promoting exercise to older adults to prevent depression, social aspects should be considered in addition to frequency	-
Karg et al. (2020) [46] The StudyKuS §, ¥	Cohort (follow-up)	To investigate the effectiveness of a manualised bouldering psychotherapy (BPT), compared with exercise alone, in a large nationwide sample of outpatients with depression.	133 (133, 100%)	18 and over 42 (SD = 12.5)	Bouldering psychotherapy &home-based exercise programmes. (programme)	Depression	MADRS (Diagnosis)	The results of the current study provide support for previous findings in suggesting positive effects of physical activity and particularly bouldering in depressed individuals. Moreover, it is evident that our bouldering psychotherapy is not only efficacious in reducing depressive symptoms but even goes beyond the benefits of mere physical exercise.	-
King et al. (2013) [47] The Longitudinal Assessment of Bariatric Surgery-2 (LABS-2) §, ¥	Cross-sectional	To examine associations between physical activity (PA) and mental health among adults undergoing bariatric surgery.	850 (735, 86.4%)	36–53 Mean not disclosed	Preoperative PA was assessed in one half of LABS-2 participants with the StepWatch™ 3 Activity Monitor (programme)	Depression	MCS and BDI (self-reported) Mild severe: score 10–18 Moderate-severe: score ≥19 Validated measure	This study revealed an inverse association between rather modest levels of PA and depressive symptoms and recent treatment for depression or anxiety, in a large cohort of adults with class 2 and 3 obesity undergoing bariatric surgery at one of 10 hospitals throughout the U.S. Although causality cannot be established, our findings are encouraging and should leverage further investigation of the role of PA in prevention and treatment of depression and anxiety in adults with class 2 and 3 obesity, as PA may prove to be a comparatively safe and cost-effective treatment option.	-
Koo and Kim (2020) [48] The KNHANES study ¥	Cross-sectional	To investigate the effects of physical activity (PA) on the stress and suicidal ideation of Korean adult women with depressive disorder.	1315 (1315, 100%)	19–65 Mean not disclosed	International Physical Activity Questionnaire (IPAQ), which is a standardized questionnaire designed to measure and compare the level of PA of various populations (aged 16–65) around the world (self-reported) Validated measure	Depression	Clinical diagnosis for depression + The perception of stress, which is a dependent variable, was asked as “How much stress do you usually feel in your daily life?” (self-reported) Low: Score = 1 High: Score = 2 Not validated measure	In this study, flexibility exercises played an important role in reducing and preventing stress and suicidal ideation in Korean adult women with depressive disorder. However, strength exercises and walking did not have significant effects on stress and suicidal ideation in Korean adult women with depressive disorder. Future studies need to consider determining which exercises aside from strength exercises, flexibility exercises, and walking are effective to reduce stress and suicidal ideation in women with depressive disorder.	-
Nam et al. (2017) [49] The KNHANES study ¥	Cross-sectional	To examine the relationship between sitting-time and MDD and estimate the effects of sitting-time and PA on MDD in a representative South Korean population.	4145 (424, 10.2%)	20 and over	Overall daily sitting time & IPAQ questionnaire (self-reported) Validated measure	Major Depressive Disorder	PHQ-9 (self-reported) score ≥ 8 Validated measure	This study showed that sitting for long periods was associated with greater risk of MDD in South Korean adults. The findings accentuated the importance of reducing overall sitting time and increasing PA and suggested that policymakers should develop strategies involving PA, to decrease sitting time and alleviate the burden of depression in terms of fiscal health premiums and social problems.	+
Pengpid and Peltzer (2019) [50] Cross-sectional data from the Global School-Based Student Health Survey (GSHS) of five Southeast Asian countries ¥	Cross-sectional	To investigate the associations of leisure-time sedentary behavior with psychological distress and with substance use among school-going adolescents in five Southeast Asian countries.	32,696 (7585, 23.1%)	11 and over (adolescents) Mean not reported Median = 14 years (IQ range = 2)	Leisure time & days per week on physical activities (self-reported) Not validated measure	Psychological distress	The psychological distress items (no, single and multiple). (self-reported) Single: score = 1 Multiple: score ≥ 2 Not validated measure	Students who spent three or more hours engaged in leisure-time sedentary behavior were more likely to have single and multiple psychological distress.	+
Shigdel et al. (2019) [32] The HUNT study ¥	Cohort (follow-up)	To examine the relationship between estimated Cardio Respiratory Fitness (eCRF) with depression and anxiety cross-sectionally and longitudinally in a representative population of middle-aged and older adults from Norway.	**Cross-sectional:** 26,615 (7141, 26.8%) **Longitudinal:** 14,020 (1847, 13.1%)	19–90 55.7 (11.4)	Two PA question on weekly duration of hard PA (being sweat and breathless) and light PA (not being sweat and breathless) from HUNT 2 (self-reported) Validated measure	Anxiety and depression	HADS-D and HADS-A (self-reported) scores ≥ 8 Validated measure	In this large cohort study, medium and high levels of eCRF were associated with a lower risk of depression as compared to those with low eCRF level, even after adjustment for well-known risk factors in both cross-sectional and longitudinal analyses. Specifically, we found 11% and 8% lower risk of depression for each unit increase in MET in cross-sectional and longitudinal data respectively. However, our data do not support a statistically significant association of MET with anxiety neither in cross-sectional analysis nor in longitudinal analysis.	- with Depression; No with Anxiety.
Steinmo et al. (2014) [51] The Whitehall II study ¥	Cohort (follow-up)	To investigate longitudinal and bidirectional associations between mental health and physical activity from midlife into old age.	6909 (1041, 15.1%)	45–69, 50–74 and 55–80 54.2 (5.7)	Total weekly hours of physical activity were converted into standardised Metabolic Equivalent of Task (MET) values (self-reported)	Probable depression & poor mental health	SF-36 MCS and GHQ (self-reported) MCS score of ≤42 GHQ ≥ 5	From midlife to old age, greater physical activity is associated with better mental health and vice versa. These findings suggest persistent longitudinal and bidirectional associations between physical activity and mental health.	-
Underwood et al. (2013) [31] RCT, no name §, Θ	Cohort (follow-up)	To test the hypothesis that a moderate intensity exercise programme would reduce the burden of depressive symptoms in residents of care homes.	765 (595, 77.7%)	65 and over Mean no disclosed	Exercise classes to provide a moderate intensity strength and aerobic training stimulus (programme)	Depression	GDS-15 (self-reported) score ≥ 15 Validated measure	This moderately intense exercise programme did not reduce depressive symptoms in residents of care homes. In this frail population, alternative strategies to manage psychological symptoms are required.	No
Van Gool et al. (2006) [52] The longitudinal Maastricht survey ¥	Cohort (follow-up)	To determine whether healthy lifestyles are associated, over time, with absence of depressed mood in the general population.	1169 (164, 14%)	21–48 48.9 (SD = 14.17)	Mean numbers of minutes spent daily on physical exercise at baseline and follow-up (self-reported) Not validated measure	Depression	CES-D (self-reported) score > 16 Validated measure	Significant longitudinal protective effect of baseline physical exercise (at recommended levels) on subsequent depressed mood.	-
Van Kim & Nelson (2013) [53] Web-based questionnaire ¥	Cross-sectional	To examine cross-sectional associations between vigorous physical activity, mental health, perceived stress, and socializing among 4-year college students.	14,706 (1145, 7.7%)	18 and over (adults) Mean no disclosed	Question from the Youth Behavior Risk survey to assess vigorous PA (self-reported) Validated measure	Poor mental health & perceived stress	The five-item mental health scale from the Short Form–36 (SF-36) health scale (self-reported) Poor MH: score <40 Perceived stress: score 9 to 16 Validated measure	In conclusion, there appears to be an inverse association between vigorous PA in college and both poor mental health and perceived stress. This relationship remained after accounting for socializing. However, additional research using longitudinal data is needed to more accurately assess the influence of PA on mental health and perceived stress from high school to college. Among college students in particular, peer support interventions aimed at either increasing or maintaining PA levels could help improve mental health and reduce perceived stress as well as maintain physical health. In addition, mental health and stress management interventions could potentially include PA components combined with social support.	-
Zhang & Yen (2015) [54] U.S. Behavioral Risk Factor Surveillance System (BRFSS) questionnaire ¥	Cross-sectional	To investigate the roles of physical activity (exercise) and sociodemographic factors in depressive symptoms among men and women in the United States.	11,560 (1056, 9.13%)	18–99 54 (16)	The measurement for PA is drawn from BRFSS questionnaire item “How many times per week or per month did you take part in this activity during the past month?” (self-reported) Not validated measure	Depression	PHQ-8 (self-reported) Severe Depression: score = 4 Validated measure	The primary finding is that regular PA ameliorates DS, decreasing the probability of moderate DS among men, and the probabilities of mild, moderate, and moderately severe DS among women. Mildly and moderately depressed women will benefit the most from regular PA. These results echo findings in previous studies, mostly with small and sectorial samples, that PA can reduce symptoms of mild to moderate depression. The use of a switching probability model allows quantification of these effects of PA and, more important, the segmented sample analysis uncovers important differences between men and women in the effects of PA on the probabilities of DS.	-

Notes. CRS: Clinically Relevant Symptoms; PA: Physical Activity; MH: Mental Health; SD: Standard Deviation; IQ range: Inter-quartile range; OR: Odds Ratio; AOR: Adjusted Odds Ratio; 95% CI: 95% Confidence Interval; *p*: *p*-value; RCT: Randomised Controlled Trial; PRO: Patient Reported Outcomes; CES-D: Center for Epidemiologic Studies Depression scale; GMS: Geriatrics Mental Status scale; IPAQ: International Physical Activity Questionnaire; VPA: Vigorous Physical Activity; MPA: Moderate Physical Activity; MVPA: Moderate-Vigorous Physical Activity; GHQ: General Health Questionnaire; BDI-II: Beck Depression Inventory scale II; PHQ: The Self-reported Patient Health Questionnaire Depression Scale; POMS: Profile of Mood States; GDS: Geriatric Depression Scale; EPDS: Edinburgh Postnatal Depression Scale; SDS: Self-rating Depression Scale; SAS: Self-rating Anxiety Scale; SCL-5: Short version of the Hopkins Symptom Check List Five-item scale; MADRS: Montgomery–Åsberg Depression Rating Scale; MCS: Mental Component Summary Score; HADS-A: Hospital Anxiety Depression Scale [Anxiety]; HADS-D: Hospital Anxiety Depression Scale [Depression]; SF-36: Short Form 36 Health Survey; BRFSS: Behavioral Risk Factor Surveillance System; BMI: Body Mass Index; CPM: Counts per minute; ST: Screen Time; BRFSS: Behavioral Risk Factor Surveillance System; MDD: Major Depressive Disorder. § Study includes as confounding factors for some type of MH treatment, either directly or indirectly; ¥ no-specification on the association of PA with MH by age groups or global analyses for the adult population only (<65). Θ Elderly papers focused on +65 population or that report specific results for this subgroup. * Population studied aged 60–101. A positive (+) overall effect represents a worsening in MH or an increase in the risk of MH due to physical inactivity. A negative (−) overall effect represents an improvement in MH or a reduction in the risk of MH as a result of PA.

**Table 2 ijerph-18-04771-t002:** Evaluation of studies according to Parmar et al. (2016) [29] Risk of Bias and quality assessment.

Study	Selection Bias	Ecological Fallacy	Confounding Bias	Reporting Bias	Time Bias	Measurement Error in Exposure (PA) Variable	Measurement Error in (Mental) Health Outcome	Overall bias Check Assessment
Annerstedt (2012) [33]	Moderate	Strong	Strong	Strong	Weak Time of data collection 2005	Moderate	Strong	Moderate
Ball et al. (2009) [24]	Moderate	Strong	Strong	Strong	Weak Time of data collection 2000–2003	Strong	Strong	Moderate
Benedetti et al. (2008) [34]	Moderate	Strong	Moderate	Moderate	Moderate	Moderate	Strong	Strong
Bishwajit et al. (2017) [35]	Strong	Strong	Moderate	Strong	Weak Time of data collection 2002–2004	Moderate	Strong	Moderate
Blumenthal et al. (2012)* [36]	Strong	Strong	Moderate	Moderate	Strong	Strong	Strong	Strong
Byeon et al. (2019) [37]	Moderate	Strong	Strong	Strong	Strong	Moderate	Moderate	Strong
Chang et al. (2020) [38]	Weak No statistical method used to predetermine sample size	Strong	Moderate	Strong	Strong	Strong	Strong	Moderate
Chang et al. (2016) [39]	Strong	Strong	Strong	Strong	Moderate	Moderate	Moderate	Strong
Chen et al. (2010) [40]	Strong	Strong	Moderate	Strong	Moderate	Strong	Strong	Strong
Coll et al. (2019)* [30]	Strong	Strong	Strong	Strong	Moderate	Strong	Strong	Strong
Feng et al. (2014) [41]	Strong	Strong	Strong	Strong	Strong	Moderate	Moderate	Strong
Griffiths et al. (2014) [42]	Strong	Strong	Strong	Moderate	Moderate	Moderate	Moderate	Strong
Guddal et al. (2019) [43]	Moderate	Strong	Strong	Strong	Weak Time of data collection 2006–2008	Strong	Strong	Moderate
Hamer et al. (2017) [18]	Strong	Strong	Moderate	Strong	Weak Time of data collection 1994–2004	Strong	Strong	Moderate
Hamer et al. (2014) [44]	Strong	Strong	Strong	Strong	Moderate	Strong	Moderate	Strong
Kanamori et al. (2018) [45]	Weak Not representative sample	Strong	Moderate	Strong	Moderate	Moderate	Strong	Moderate
Karg et al. (2020) [46]	Strong	Strong	Strong	Strong	Strong	Strong	Strong	Strong
King et al. (2013) [47]	Strong	Strong	Strong	Strong	Moderate	Strong	Strong	Strong
Koo and Kim (2020) [48]	Strong	Strong	Moderate	Strong	Moderate	Strong	Weak Not validated scale	Moderate
Nam et al. (2017) [49]	Strong	Strong	Moderate	Strong	Moderate	Strong	Strong	Strong
Pengpid and Peltzer (2019) [50]	Strong	Strong	Strong	Weak Inaccurate reporting some aspects of the study	Moderate	Moderate	Moderate	Moderate
Shigdel et al. (2019) [32]	Strong	Strong	Strong	Strong	Moderate	Strong	Moderate	Strong
Steinmo et al. (2014) [51]	Weak	Strong	Moderate	Strong	Moderate	Moderate	Moderate	Moderate
Underwood et al. (2013) [31]	Strong	Strong	Strong	Strong	Moderate	Moderate	Moderate	Strong
Van Gool et al. (2007) [52]	Moderate	Strong	Moderate	Strong	Weak	Strong	Strong	Moderate
VanKim & Nelson (2013) [53]	Strong	Strong	Moderate	Weak Inaccurate reporting some aspects of the study	Moderate	Strong	Strong	Moderate
Zhang & Yen (2015) [54]	Moderate	Strong	Strong	Strong	Strong	Moderate	Strong	Strong

* Study is a RCT reporting good sampling methods and appropriate design.

## Data Availability

Not applicable.

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
