# Peer review of "The Association of Physical (in)Activity with Mental Health. Differences between Elder and Younger Populations: A Systematic Literature Review"

_ijerph, 2021, doi:10.3390/ijerph18094771_

Round 1
Reviewer 1 Report
First of all, it strikes me as a reviewer and specialist doctor in the field, psychophysiologist expert, that the three authors are from the area of ​​economics. And that regarding the exposed subject, there is no expert.
Remove the numbering in the abstract.
Parentheses must be replaced by square brackets.
At this point, I have too many doubts about the authorship of the text... the format and the type of writing.
1
LCF / PR / CE07 / 50610001 belongs to grants and has been used in biomedical research in which the authors of the present manuscript do not appear.
The money from the financing of Secretaria d’Universitats i Recerca del Departament d'Economia i Coneixement de la Generalitat de Catalunya, has it been used in research for a completely disparate area of ​​knowledge, and to carry out a bibliographic review work??? Really??
Marlene Herisson and Marc Saez are the original authors of the article?
Author Response
Although we appreciate the time you devoted to read and review the paper, we have to disagree with some of your comments. First of all, we are all health economists with a solid background on healthy lifestyle and mental illness. We are confident that we have enough expertise to run a rigorous systematic review and present concluding results. Note that health economists do not only perform economic evaluation. In fact, none of us does it. Our expertise in health economics make us comfortable with a basic medical vocabulary. In addition, the subject of the manuscript was not too technical, so we did not need to include a medical / clinical co-author. In addition, we provided (in the submission) all the material used for the search strategy and the tables with all the papers included and excluded from the analysis. Aside from this material and the abovementioned claims, unfortunately, there is not much we can do to dissipate your doubts given this is a blinded process. Otherwise, we would be happy to share our CVs, where you could check that we are three hard-working researchers, with a reputation in the field.
Regarding your mention to Marlene Herisson and Marc Saez:
On May 2018, two of the authors decided to continue our research on mental health[1] looking at the role of lifestyle. By that time Marlene Herisson (a master student) visited our institution and given our common interest we decided to focus on the relationship between mental health and physical activity. Herisson developed her master thesis (a systematic literature review on the association of physical activity with mental health), directed by us; and we learned more insights on the topic to develop an empirical paper. Marc Saez was involved in the interpretation of the results of that thesis. Given the outstanding quality of the thesis, we published a working paper to share our knowledge. However, the contribution was still modest in order to be published in a leading journal.
Given that our main field of research involved the elderly, we strongly believed that we could contribute with a new systematic review on the topic exploring heterogenous effects among adult and elderly populations. Indeed, we got the idea after working on an empirical paper where we checked on the effects of a physical activity intervention on a relatively elder population (which we comment below to justify the funding received). We therefore contacted another colleague, to have the view of a new and fresh pair of eyes, with experience in systematic literature reviews in health economics, as well as with experience in mental health research. The three authors worked together and agreed with a new search strategy (the one implemented in the manuscript you revised) that ended up as a completely different search strategy compared to the one that was used in the working paper version with Herisson and Saez. The new search strategy was definitely more rigorous methodologically. We also introduced an additional perspective of the analysis and focus on the comparison between younger and elder populations, as we saw a literature gap in this sense. To sum up, accepting to proceed with a new search strategy and scope and exploring heterogeneous effects by age required re-doing the review again completely from scratch and re-writing the paper. As Herison and Saez were not involved, they are not co-authoring the paper.
We are sorry but we are not sure how to respond to your comment about the format and writing style, given that you are not providing any clues for us to review whatever was that shocked you. We can only say that the language of paper was reviewed by an English native person before submission. Indeed, that is the reason why one authors acknowledge financial support to AGAUR 2017 SGR 1059 grant, issued by “Secretaria d’Universitats i Recerca del Departament d'Economia i Coneixement de la Generalitat de Catalunya” (used to pay the English revision fee). Despite being a minor financial support, we feel indebted to report it.
In terms of funding, this research is part of a bigger project funded by the Center of Research in Health and Economics (CRES – Universtitat de Barcelona) through the funding received by Fundació Bancaria Caixa d'Estalvis i Pensions de Barcelona: agreement LCF/PR/CE07/50610001. The recipient of the funding is the CRES research centre (i.e., Guillem López Casasnovasm as director) and then CRES funds this project in an agreement with Fundació Bancaria Caixa d'Estalvis i Pensions de Barcelona. If you require a proof to validate that, we can submit it. The topic of this big project is the evaluation of a policy implemented in the Catalan health system to encourage the elder population with chronic conditions (i.e., cardiovascular) to engage in physical activity. Our aim is to see the effect of this policy on mental health. We work with survey data. However, we decided that a systematic review on the topic focusing on ageing was needed in order to have all the information before starting with the policy evaluation study. So, given that the project indirectly also supports financially this paper we felt that we had to acknowledge the financial support too.
Regarding your two only minor comments to improve the paper:
- We have removed the numbering in the abstract.
- We also have replaced, for the reference’s citation, parentheses by square brackets in all the manuscript.
Footnotes:
[1] Our research on mental health, so far focused on ill-patients suffering from mental health.
Reviewer 2 Report
The article is about a systematic review study of the association of physical activity and mental health. While this topic seems to be of importance, there are a few revisions that I recommend before accepting this document for publication. The following are section-specific questions, comments, and concerns.
Introduction
The introduction is not very consistent, there is a lack of more data on the mental health of the general population and on physical activity or inactivity that focus on the problem to be studied. For example, the percentage of people who suffer from a mental health problem at the European level. It is recommended to review the report of the EU Joint Action on Mental Health and Weebeing (2016).
line 72-79 this is part of the results.
line 83-93 It is part of the study procedure (Method)
Discussion.
Line 2-8 mentions three specific objectives, in the introduction only two specific objectives are mentioned (line 66-70).
Part of the results are presented in the discussion and are not discussed as such.
line 10 to 68, is not discussion is result.
A more exhaustive discussion of the results remains to be done on the effects that physical activity (frequency, intensity, type of exercise) has on mental illnesses (type of illness and age of the subjects).
line 21, Capital letter “below”
Line 162-168, the objective is repeated.
Author Response
The article is about a systematic review study of the association of physical activity and mental health. While this topic seems to be of importance, there are a few revisions that I recommend before accepting this document for publication. The following are section-specific questions, comments, and concerns.
Response: Thank you for your comment. We agree with you on that the topic is policy relevant and appreciate your comments that really improved the quality and clarity of the manuscript. We now respond one by one to each of your comments. At the end of our response, we will also include some additional changes that were necessary (as a result of your comments and of the comments of other reviewers) for keeping the consistency of the text.
Note that, even though we left the track changes visible, when we refer to specific lines in the text, we are using the “Simple view” format.
Introduction
Comment 1. The introduction is not very consistent, there is a lack of more data on the mental health of the general population and on physical activity or inactivity that focus on the problem to be studied. For example, the percentage of people who suffer from a mental health problem at the European level. It is recommended to review the report of the EU Joint Action on Mental Health and Weebeing (2016).
Response: Thanks for this comment. We have added some data on prevalence according to the last published report of the EU Joint Action on Mental Health and Wellbeing (2016), by Barbato et al. (2016).
The added text is in lines 33-35: “The latest European Union (EU) report, in the frame of the EU Health Programme 2014-2020, suggest an overall estimate of one-year prevalence of any mental health disorder around 38% [2]”
Reference [2], Barbato et al. (2016), has also been added to the references’ list at the end of the document.
Comment 2: line 72-79 this is part of the results.
Response: Thank you. We have removed the following text from the introduction section: “We extracted and analysed information from 27 studies that analysed the association of PA with MH from an empirical perspective. We found, overall, an inverse association of PA with MH, showing that higher levels of PA are related to a reduction in adverse MH outcomes.”
Comment 3. line 83-93 It is part of the study procedure (Method)
Response: Thank you. We have removed the following text from the introduction section: “We consider objective and subjective measures for both PA and MH. Objective measures of PA are those recorded by an external technology (e.g. accelerometer recording number of steps or time performing the exercise) or by an exercise supervisor (e.g. coach). Manually recorded PA by the individual, for example, through a questionnaire or interview, is considered a self-reported type of PA. Regarding MH, measures are considered self-reported MH when they are not measured through a medical diagnosis. Only medical diagnoses are considered objective measures of MH. Subjective measures considered for PA and MH can be distinguished between those measured based on validated scales (e.g IPAQ questionnaire, the only validated scale found for PA in this review[1], or GHQ-12 or PHQ-8, amongst others, for MH) and non-validated scales (e.g., questions for PA like “How often are you physically active of perform exercise during your leisure time? Excluding domestic work” where response alternatives offered are: 1) Sedentary 2) Moderate physical activity 3) Regular exercise 4) Regular advanced exercise), and questions for MH like “Have you ever been diagnosed with depression?”).”
This paragraph has been added (and adapted) to the methods section, specifically at the end of section 2.3 Eligibility criteria (lines 143-171, at page 5). The text has been slightly reviewed. Our revised text is below:
“We consider objective and subjective measures for both PA and MH (if subjective, only clinically relevant measures). Objective measures of PA are those recorded by an external technology (e.g. accelerometer recording number of steps or time performing the exercise) or by an exercise supervisor (e.g. coach). Manually recorded PA by the individual, for example, through a questionnaire or interview, is considered a self-reported type of PA. Regarding MH, measures are considered self-reported MH when they are not measured through a medical diagnosis. Only medical diagnoses are considered objective measures of MH. Subjective measures considered for PA and MH can be distinguished between those measured based on validated scales (e.g. IPAQ questionnaire, the only validated scale found for PA in this review[2], or GHQ-12 or PHQ-8, amongst others, for MH) and non-validated scales (e.g., questions for PA like “How often are you physically active of perform exercise during your leisure time? Excluding domestic work” where response alternatives offered are: 1) Sedentary 2) Moderate physical activity 3) Regular exercise 4) Regular advanced exercise), and questions for MH like “Have you ever been diagnosed with depression?”
Yet, we have added a shorten version of this linked to the contribution of the paper, as it was suggested by another reviewer.
In lines 74-92 at the Introduction section (pages 2-3): “Hence, the main contribution of this review is twofold. First, we look for heterogenous effects in the literature by age (below and above 65 years old). Second, we distinguish between objective and subjective measures of both, PA and MH; moreover, subjective self-reported measures are distinguished in case of validated scales. In addition, we have also identified that previous reviews lead to weak findings because they include papers based on (i) clinically irrelevant MH problems and (ii) descriptive non-robust statistical analysis. Thus, our goal is achieved by conducting a systematic literature review, focusing on studies conducting some type of econometric analysis, excluding studies that are purely descriptive, and selecting evidence based on clinically relevant MH problems[3]. As summarized in Figure 1, weak evidence is excluded, minimizing the risk of biased results. Imposing strong inclusion criteria (clinical characteristics and methodological restrictions) ensures better comparability among the selected studies, guaranteeing the robustness of our findings. On the one hand, our selection criteria make more likely that the people self-reporting MH problems to resemble to clinically diagnosed patients than if we had included all those other papers that use self-reported MH measures score as a continuum, or that do not use a cut-off score. On the other hand, the restriction to papers using econometric methods restrict the included papers to those that provide information that can be used to establish an association of PA with MH, something that would not be possible using papers conducting purely descriptive analyses.”.
Discussion
Comment 4. Discussion. Line 2-8 mentions three specific objectives, in the introduction only two specific objectives are mentioned (line 66-70).
Response: Thank you for your comment. We have rewritten the part of the objectives’ description at the introduction section. The text (now in lines 68-73, page 2) is now consistent with the text at the discussion section: “Specific objectives are to assess whether: i) there are differences in the association of PA with MH between the elder and younger populations; ii) there are differences in the association of PA with MH according to the type of PA measured (objective vs. subjective); and iii) there are differences in the association of PA with MH according to the type of MH measured (objective vs. subjective)– with a focus on clinically relevant symptoms when MH is subjective.”
Comment 5. Part of the results are presented in the discussion and are not discussed as such.
Response: Thank you for your comment. We followed the format of a previous and recently published paper (Jedlinski et al. (2021)) in the journal. We observed that, in this paper, the main results are presented in the discussion section. However, having seen this comment, we have moved most of the content in the discussion subsections to the results section.
Specifically, sections 4.1, 4.2, and 4.3 have been moved to section 3.3 Main results. The new sections are:
- 3.1 Differences in the association of PA with MH between elder and younger populations
- 3.2 Differences in the association of PA with MH between self-reported and objective types of MH
- 3.3. Differences in the association of PA with MH for self-reported and objective types of PA.
The text of previous Section 4.4, referring to limitations and contribution of this work, was kept for the discussion section. We have, though, rewritten the whole discussion section as you can see the new manuscript.
Comment 6. line 10 to 68, is not discussion is result.
Response: Thank you. This has been answered as was related with comment 5.
Comment 7. A more exhaustive discussion of the results remains to be done on the effects that physical activity (frequency, intensity, type of exercise) has on mental illnesses (type of illness and age of the subjects).
Response: Thank you. The whole section has been reviewed. To deal with this suggestion, we wrote a new paragraph in lines 184-193: “Indeed, if we compare the results observed in those studies using patients clinically diagnosed of MH with results from studies using self-reported clinically relevant MH measures, we observe similar findings. In particular, the two studies using mentally-ill populations and twenty-one out of the twenty-five studies using self-reported clinically relevant MH measures found a negative and unconditional association of PA with MH, indicating PA is beneficial for MH (and (in)PA worsens MH). The remaining four studies using self-reported MH measures found a conditional association, such as, for example, that some therapies would work better than others.”
Comment 7. line 21, Capital letter “below”
Response: Thank you. Due to comments of another reviewer, who suggested to create one single table combining tables 1 and 2 (in the original submitted version of the document), the sentence has been changed to (lines 14-15): “We identified in Table 1 the number of studies that used objective or self-reported, and validated vs. not validated, measures of PA and MH.”
Comment 8. Line 162-168, the objective is repeated.
Response: Thank you. The following text has been removed: “Our review focused on the association of PA with MH, but we went one step beyond analysing the distinction between objective and self-reported measures of both PA and MH.”
Additional changes
Due to comments from other reviewers, you will notice there are some additional changes (besides the changes that result from your own review). In short, we have added one paragraph about our contribution and Figure 1 in the introduction, and we have combined tables 1 and 2.
References
JedliÅ„ski, M., Mazur, M., Grocholewicz, K., & Janiszewska-Olszowska, J. (2021). 3D Scanners in Orthodontics—Current Knowledge and Future Perspectives—A Systematic Review. International Journal of Environmental Research and Public Health, 18(3), 1121.
Footnotes:
[1] Note that there exist other validated scales for measuring objective PA, such as the GPAQ questionnaire. However, there were no studies using the GPAQ questionnaire that satisfied the inclusion criteria specified for the objective of this review.
[2] Note that there exist other validated scales for measuring objective PA, such as the GPAQ questionnaire. However, there were no studies using the GPAQ questionnaire that satisfied the inclusion criteria specified for the objective of this review.
[3] A clinically relevant MH problem is defined by validated score cut-offs for certain instruments for the measurement of self-reported MH problems; e.g. a score over 10 points in the CES-D questionnaire is used as an indicator of clinically relevant depression symptoms in Ball et al. [24], according to a previous validation study [25].
Reviewer 3 Report
Authors have performed significant amount of literature review to examine the effects of physical activity on cognitive function in different age populations. although well performed literature review, lack of novelty is the major issue for this article. additionally, table 1, 2, and 3 should be combined together to avoid any confusions for the readers. furthermore, I strongly suggest that authors should provide a schematic figure to summarize their hypotheses... this figure should be figure 1 or 2. then readers may be able to expect and understand what the authors try to suggest in this review paper.
Author Response
Authors have performed significant amount of literature review to examine the effects of physical activity on cognitive function in different age populations. although well performed literature review, lack of novelty is the major issue for this article. additionally, table 1, 2, and 3 should be combined together to avoid any confusions for the readers. furthermore, I strongly suggest that authors should provide a schematic figure to summarize their hypotheses... this figure should be figure 1 or 2. then readers may be able to expect and understand what the authors try to suggest in this review paper.
Response.
First of all, thank you for the time you devoted to review our article, and for all your comments.
Please, note that even though we left the track changes visible, when we refer to specific lines in the text, we are using the “Simple view” format.
We agree with you that a systematic review between physical activity (PA) and mental health (MH) is not novel. However, our contribution and the novelty of the manuscript/research is twofold. First, we look for heterogenous effects in the literature by age (below and above 65 years old). And second, we distinguish between objective and subjective measures of both, PA and MH; moreover, subjective or self-reported measures are distinguished between validated scales and other type of variables. We have added the following text to make clearer our contribution at the introduction section (that we believe it is clearer now):
In lines 74-92, pages 2-3: “Hence, the main contribution of this review is twofold. First, we look for heterogenous effects in the literature by age (below and above 65 years old). Second, we distinguish between objective and subjective measures of both, PA and MH; moreover, subjective self-reported measures are distinguished in case of validated scales. In addition, we have also identified that previous reviews lead to weak findings because they include papers based on (i) clinically irrelevant MH problems and (ii) descriptive non-robust statistical analysis. Thus, our goal is achieved by conducting a systematic literature review, focusing on studies conducting some type of econometric analysis, excluding studies that are purely descriptive, and selecting evidence based on clinically relevant MH problems[1]. As summarized in Figure 1, weak evidence is excluded, minimizing the risk of biased results. Imposing strong inclusion criteria (clinical characteristics and methodological restrictions) ensures better comparability among the selected studies, guaranteeing the robustness of our findings. On the one hand, our selection criteria make more likely that the people self-reporting MH problems to resemble to clinically diagnosed patients than if we had included all those other papers that use self-reported MH measures score as a continuum, or that do not use a cut-off score. On the other hand, the restriction to papers using econometric methods restrict the included papers to those that provide information that can be used to establish an association of PA with MH, something that would not be possible using papers conducting purely descriptive analyses.”.
The text directly links to a new Figure, Figure 1 which aims to better summarise our hypothesis. Existing systematic review pool evidence without accounting for quality (mainly based on pure and simple descriptive analysis which does not identify clinically relevant MH problems). Hence, previous systematic reviews include more papers, but with weak results. Our hypothesis is that being more restrictive in papers selection leads to stronger, more precise and robust findings.
Consequently, The PRISMA figure has been renamed as Figure 2. In pages 4 (line 131) and 5, (line 152), the text has changed and is now “Figure 2”.
Regarding the table comment, we have merged tables 1 and 2 into a single table. We also included some additional explanations.
- First, we have added two columns (Study objectives and Summary of results).
- Second, columns referring to the MH and PA outcome measures, include now a comment on the validity of the measure (Validated / Non validated measure).
- Finally, in the first column of the table, you will notice we have added the following symbols to clarify some important papers’ characteristics (explanations are in the note of the table: “§ Study includes as confounding factors for some type of MH treatment, either directly or indirectly; ¥ no-specification on the association of PA with MH by age groups or global analyses for the adult population only (<65). Θ Elderly papers focused on +65 population or that report specific results for this subgroup. *Population studied aged 60-101.”)
Regarding merging the content of table 3, we prefer to leave it as is, for clarity purposes. We believe this change facilitates the flow of the results section.
Additional changes
Due to comments from other reviewers, you will notice there are some additional changes (besides the changes that result from your own review). In short, we have moved one paragraph of the introduction to methodology, and some section of discussion to results. In addition, discussion has been rewriting emphasising the contribution. In the introduction, we have also added more statistical evidence and re-write the objective part to be more consistent.
Footnotes
[1] A clinically relevant MH problem is defined by validated score cut-offs for certain instruments for the measurement of self-reported MH problems; e.g. a score over 10 points in the CES-D questionnaire is used as an indicator of clinically relevant depression symptoms in Ball et al. [24], according to a previous validation study [25].
Round 2
Reviewer 1 Report
.
Author Response
Thank you for your devoted time and revisions. Please, note that although we cleaned the document of track changes, we clearly indicated the changes using comments along the text. In addition, the manuscript language and style has been reviewed thoroughly.
Reviewer 2 Report
The authors made extensive changes to the manuscript based on reviews. Here are some minor changes that are needed.
Footnote 1 I do not understand why the term Physical activity is clarified, since this is understood as any energy expenditure and is different from the definition of sport or physical exercise. Maybe it should be within the introduction itself and not in a footnote
“1Physical activity (PA) does not only include sports and active forms of recreation (e.g., dancing), but also refers to mobility (walking and cycling), work-related activities and household tasks [5]. Numerous health organisations (CDC, WHO, Health and Human Services) have outlined the benefits of physical activity, including a reduction in mental health risks”.
Footnote 2 Similar footnote 1
Line 98- 101 This argument should be in the discussion
“Our review suggests that more evidence is needed regarding the effect of ageing when measuring associations of physical activities with mental health, and that there is a lack of studies analysing this association in people with probable MH, ignoring the similarity between this population and the population with a clinical diagnosis on MH”
Line 196- 197. “Although our review suggests evidence in favour of a negative relationship between PA levels with MH problems”
I need a clarification; in the previous paragraph they indicate that physical activity is beneficial for mental health.
Do you mean that the less physical activity there are more health problems?
Author Response
The authors made extensive changes to the manuscript based on reviews. Here are some minor changes that are needed.
Thank you for your devoted time and revisions. Please, note that although we cleaned the document of track changes, we clearly indicated the changes using comments along the text. In addition, the manuscript language and style has been reviewed thoroughly.
Footnote 1 I do not understand why the term Physical activity is clarified, since this is understood as any energy expenditure and is different from the definition of sport or physical exercise. Maybe it should be within the introduction itself and not in a footnote
“Physical activity (PA) does not only include sports and active forms of recreation (e.g., dancing), but also refers to mobility (walking and cycling), work-related activities and household tasks [5]. Numerous health organisations (CDC, WHO, Health and Human Services) have outlined the benefits of physical activity, including a reduction in mental health risks”.
Thank you for this comment. We have reinserted these two sentences in in the text. Note that due to the English editing, the text has slightly changed, but the meaning is the same.
Lines 45-47: “Physical activity (PA) does not only include sports and active forms of recreation (e.g., dancing), but also refers to mobility (walking and cycling), work-related activities and household tasks [5].”
Lines 48-50: “Numerous health organisations (CDC, WHO, Health and Human Services) have outlined the benefits of physical activity, including a reduction in the risk of suffering mental health problems.”
Footnote 2 Similar footnote 1
Thank you for this comment. We have reinserted in the text (lines 88-91 in the manuscript) the following text:
“A clinically relevant MH problem is defined by validated score cut-offs for certain instruments used in the measurement of self-reported MH problems; e.g. a score over 10 points in the CES-D questionnaire is used as an indicator of clinically relevant depression symptoms in Ball et al. [24], according to a previous validation study [25]”.
Line 98- 101 This argument should be in the discussion
“Our review suggests that more evidence is needed regarding the effect of ageing when measuring associations of physical activities with mental health, and that there is a lack of studies analysing this association in people with probable MH, ignoring the similarity between this population and the population with a clinical diagnosis on MH”
Thank you for this comment. We have removed the mentioned text from the introduction. We have inserted it in lines 202-206 at the discussion section. We have slightly modified the sentence for a better flow of the text.
Line 196- 197. “Although our review suggests evidence in favour of a negative relationship between PA levels with MH problems”
I need a clarification; in the previous paragraph they indicate that physical activity is beneficial for mental health.
Do you mean that the less physical activity there are more health problems?
Thank you for this comment. We understand the confusion. Having mental health problems is a negative. Therefore, a negative relationship between Physical Activity and mental health means mental health problems are reduced by increased physical activity. We have rephrased to clarify this point.
Lines 195-198 are now:
“Although our review provides evidence indicating PA is beneficial for MH, we observe that the intensity of such a relationship varies by the type of PA and MH measured, as well as by age.”
Reviewer 3 Report
Manuscript has been significantly improved. I suggest that this manuscript should get accepted after English editing service.
Author Response
Thank you very much for this comment.
We have incorporated some minor changes in the text, given the comments from other reviewers, and the Manuscript has been sent for a final English editing service.